# An assessment of organisational culture in Australian hospitals using employee online reviews

Antoinette Pavithra *, Johanna Westbrook

Australian Institute of Health Innovation, Centre for Health Systems and Safety Research, Macquarie University, Sydney, Australia

* Antoinette.pavithra@mq.edu.au

**Data Availability Statement:** All relevant data are within the paper. Underlying textual content from third party websites (Seek, Indeed and Glassdoor) cannot be reproduced because of limitations within website terms and conditions. "Textual content"

## Abstract

The aim of this study was to analyse the content of online reviews posted by hospital employees on job websites for themes of organisational culture. 103 anonymous online reviews across three job websites, posted by hospital employees of four hospitals within an Australian health network were extracted. Reviews had been posted across a period of six years, from 2014 to early 2020. Sentiment rating based on user-allotted ratings on the selected websites was calculated. The sentiment score was validated against the textual content of the review to confirm employee recommendation and sentiment. Sentiment was coded as neutral, positive, or negative. Significant keywords, associations, and usage within the context of identified sentiment were deductively coded and clustered manually against eight pre-determined safety culture themes. These themes were derived from the most used assessment tools for self-reported measures of occupational health and safety, and safety culture in healthcare. Workers across clinical roles (n = 49) and non-clinical roles (n = 50) were evenly represented in the dataset. 35.9% of commenters (n = 37) reported their length of employment in the hospitals that they reviewed. Most online employee reviews addressed broad themes related to perceptions of management (n = 98), safety climate (n = 97), teamwork climate (n = 91) and working conditions (n = 98). A significant set of reviews addressed themes related to job satisfaction (n = 49) and learning, training, and development (n = 41). 72.8% of online reviews (n = 75) expressed positive sentiment towards their employer. Reviews expressing negative sentiment were largely posted by former employees and indicated areas of discontent that reflected organisational and systemic factors. Online employee reviews posted by hospital workers on job sites provide valuable insights into healthcare organisational culture. Therefore, employee online reviews could be used as a supplementary source of data to inform organisational employee engagement initiatives.

## Introduction

Organisational culture within Australian healthcare organisations is typically measured using surveys. Multiple survey instruments exist and seek to measure employee satisfaction,

refers to the entirety of the online reviews posted by website users. Permission for the analysis of this content, and the publication of research findings was sought and granted by the service providers where applicable. All content used in this analysis is available directly on Glassdoor, Seek and Indeed.

**Funding:** This research was supported by the Research Training Program (RTP) scholarship allocation number 2018378/ 20191504 received by AP. The scholarship is offered by the Australian Government and administered by Macquarie University. The funder had no role in study design, data collection and analysis, decision to publish, or preparation of the manuscript.

**Competing interests:** The authors have declared that no competing interests exist.

engagement, and organisational culture. While these assessment tools generate detailed insights into employee perceptions and experience, administering surveys, analysing results, and applying the insights to achieve change within organisations can be resource and time intensive. Further, organisational culture surveys often fail to measure the occupational (OHS) or workplace health and safety (WHS) climate for healthcare workers [1]. WHS refers to the management of risks to the health and safety of all human stakeholders associated with the business conducted by an organisation [2]. Organisational culture in the context of an organisations' employees is described as the composite of the shared and common values, attitudes, beliefs and behaviours that workers espouse and display by adhering to the formal and informal rules that operate within their workplace [3]. Over the last two years, Australian legislation surrounding workplace health and safety has been reformed to account for psychosocial risk factors, therefore making organisations responsible for providing a psychologically and socially safe workplace and culture to avoid causing harm or injury to any stakeholders who interact with the organisation [4]. While patient safety and occupational safety climate are positively related, existing instruments do not concomitantly and rigorously measure patient as well as staff health and safety within hospital settings [5]. This differential focus for multiple sets of stakeholders poses a problem to accurately assessing organisational culture, workplace safety and its impacts on organisational and staff and patient outcomes, within the context of healthcare. The broad themes identified by existing instruments that measure staff perceptions and experiences of organisational culture and workplace safety such as the Safety Attitudes Questionnaire measure organisational climate categorised under the following concepts: Perceptions of Management, Risk Management, Safety Climate, Teamwork Climate, Working Conditions, Stress Recognition, Job Satisfaction and Learning, Training and Development [6, 7]. In the absence of a composite scale to assess workplace safety and organisational culture within healthcare for multiple stakeholders associated with an organisation, the proliferation of data contained within online reviews by patients and staff about their experiences, either as patients or as staff may provide valuable information to understand themes of safety and culture. Therefore, indicators of workplace safety and improvements to healthcare organisational culture can potentially be informed by myriad tools, including online reviews [8]. Online reviews have been associated with the potential of increased issue of liability and have therefore been viewed with apprehension by organisations and health practitioners [9]. Yet online reviews are increasingly being used by employees during the process of job-seeking to identify suitable employers and understand organisational behaviour [10]. Online job websites are also used by recruiters and organisations to identify and recruit suitable candidates who list their professional profiles to be visible to potential employers. These websites offer jobseekers the option of remaining anonymous when they post reviews about their workplaces, and reviews cannot be linked back to individual users' profiles. Online employee reviews have been demonstrated as a valuable source of information for highlighting areas for organisational and operational improvements [11, 12]. No studies have investigated the extent to which online job site reviews reflect organisational culture as measured through more systematic measures such as large-scale organisational surveys of staff. Our study aimed to assess the aspects of organisational culture reflected in employee online reviews posted about a network of Australian hospitals.

## Methods

An Australian health network with multiple hospital sites was selected for this study because it was undertaking a culture change intervention across its member hospital sites. The study was designed to establish a baseline of employee sentiment across the selected hospital network

using a range of online job sites. Four of the hospitals within this network are represented within three online job sites: Indeed, Glassdoor and Seek. These websites were chosen as they offered the option of reviewing workplaces to employees and had the most well-developed scales to collate anonymous ratings. No other job and career websites in Australia had any more reviews about the hospital network in the study at the time of conducting this study. A search was run for the name of the hospital network being studies on each of the three websites, and all reviews available that were related to any of the hospitals within this network in Australia were included in the study. In total, 103 online reviews were extracted from these online sites for the period of 2014 to April 2020. These were all the available reviews posted by users that were intelligible, referred to the right organisation and had been moderated and posted by the websites listed above. The characteristics of reviewers who posted the online reviews were determined through a combination of self-reported job titles and narrative descriptions within reviews of the roles undertaken during employment.

The three online job sites offer reviewers the option of selecting ratings across the broad themes listed in Table 1, namely, work-life balance, salary and benefits, career development, management, and culture. The rating system offered to users across all sites allowed each theme to be scored on a five-point Likert scale. The average rating provided by each reviewer was calculated against the maximum score allocated by each site to its themes (Indeed– 25, Glassdoor– 25 and Seek– 30). Overall sentiment ratings were calculated by converting each review's site-based rating into standardised percentages.

Reviewers are requested to provide sentiment ratings for each theme, and we calculated an average percentage score for each review. Reviewers are also asked to rate whether they would recommend this employer using a dichotomous two-point rating scale (yes / no), but not all reviewers completed this item. One site (Indeed) did not offer the option of recommendations. 42 (41.2%) out of the 102 usable reviews across Glassdoor and Seek provided either positive or negative recommendations. The sentiment and employer recommendation scores were compared against the textual content of the review to determine consistency with the reviewer sentiment score provided. When narrative content conflicted with the sentiment rating, the narrative sentiment expressed by the reviewer was given precedence, which occurred in only one case. All reviews were then given an overall sentiment score coded as neutral, positive, or negative based on a combination of the sentiment and employer recommendation rating (Table 2).

Reviewer sentiment was coded as positive if the average rating was 60 or above and the reviewer had given the employer a positive "recommendation" where the field was available. Positive sentiment coding was retained in the absence of an employer recommendation if the sentiment rating was 60 or above. Neutral sentiment was coded for ratings between 50 and 60. Where reviewers had provided a negative employer recommendation, sentiment was coded as negative irrespective of sentiment rating. All sentiment ratings below 50 were coded as negative.

**Table 1. Employee rating themes and rating scales across three sites.**

| Indeed | Glassdoor | Seek |
|---|---|---|
| Work / Life Balance (5) | Work / Life Balance (5) | Work / Life Balance (5) |
| Salary / Benefits (5) | Compensation and Benefits (5) | Benefits and Perks (5) |
| Job Security / Advancement (5) | Career Opportunities (5) | Career Development (5) |
| Management (5) | Senior Management (5) | Management (5) |
| Job Culture (5) | Culture and Values (5) | Working Environment (5) |
| | | Diversity and Equal Opportunity (5) |
| **Percentage calculated against a total of 25 points** | **Percentage calculated against a total of 25 points** | **Percentage calculated against a total of 30 points** |

**Table 2. Criteria for assigning overall sentiment score.**

| Positive | Neutral | Negative |
|---|---|---|
| Average sentiment score ≥60 and Employer recommendation positive, or no recommendation provided | Average sentiment score≥50 and <60 and no negative recommendation provided | Employer recommendation score negative, regardless of sentiment score OR where the sentiment score <50 |

Each of the narrative reviews was examined to identify keywords, associations between terms, and usage within the context of the entire review to identify overall sentiment. Simple automated analytical methods such as keyword counting cannot be applied to understand the overall sentiment of these reviews as complex contextual information is often provided. Keywords such as "challenge" or "performance" could be used with positive, negative, or neutral connotations. Therefore, it was imperative to perform a thematic analysis and clustering manually on the textual content of the reviews. Reviews were categorised and clustered against eight pre-determined safety culture themes. These themes were derived from the most used assessment tools for self-reported measures of occupational health and safety and safety culture across the network of hospitals in this study [6]. These themes were: Perceptions of Management, Risk Management, Safety Climate, Teamwork Climate, Working Conditions, Stress Recognition, Job Satisfaction, and Learning, Training and Development. Fig 1 visually represents the research methodology employed by this study.

### Ethics

Ethics approval was required for the study and was granted by the Macquarie University Medicine & Health Sciences Ethics Review Subcommittee on 21/02/2020. As all reviews analysed were anonymous, it was not possible to seek individual consent of employees who posted the online reviews. Therefore, the study was limited to thematic aggregation and no content from the reviews have been reproduced in any form. Permission for analysis of the reviews using our research method was sought and granted from the online job websites where the reviews were posted.

### Results

Online reviewers on the job sites examined in our study addressed a range of themes related to the Australian hospital network that employed them. Staff who posted online reviews appeared to use these spaces in a similar manner to staff experience surveys–to provide positive and negative feedback or narrate their workplace experiences. Across the websites, users are asked to provide overall ratings for similar types of categories which cover the themes of work-life balance, compensation and benefits, career and advancement opportunities, perceptions about management and organisational culture and values. The orientation of the unstructured textual content of the reviews appeared to often be guided by the themes that the site had surveyed users on–even though the open text field was not accompanied by specific instructions limiting the thematic breadth of the review.

### Reviewer characteristics

35.9% of commenters (n = 37) had indicated their length of employment across the hospitals that they had reviewed (Tables 3 and 4).

Reviewers were evenly split in terms of those employed in clinical roles (n = 49) and non-clinical roles (n = 50) (Table 5). No other characteristics could be determined from the reviews with accuracy.

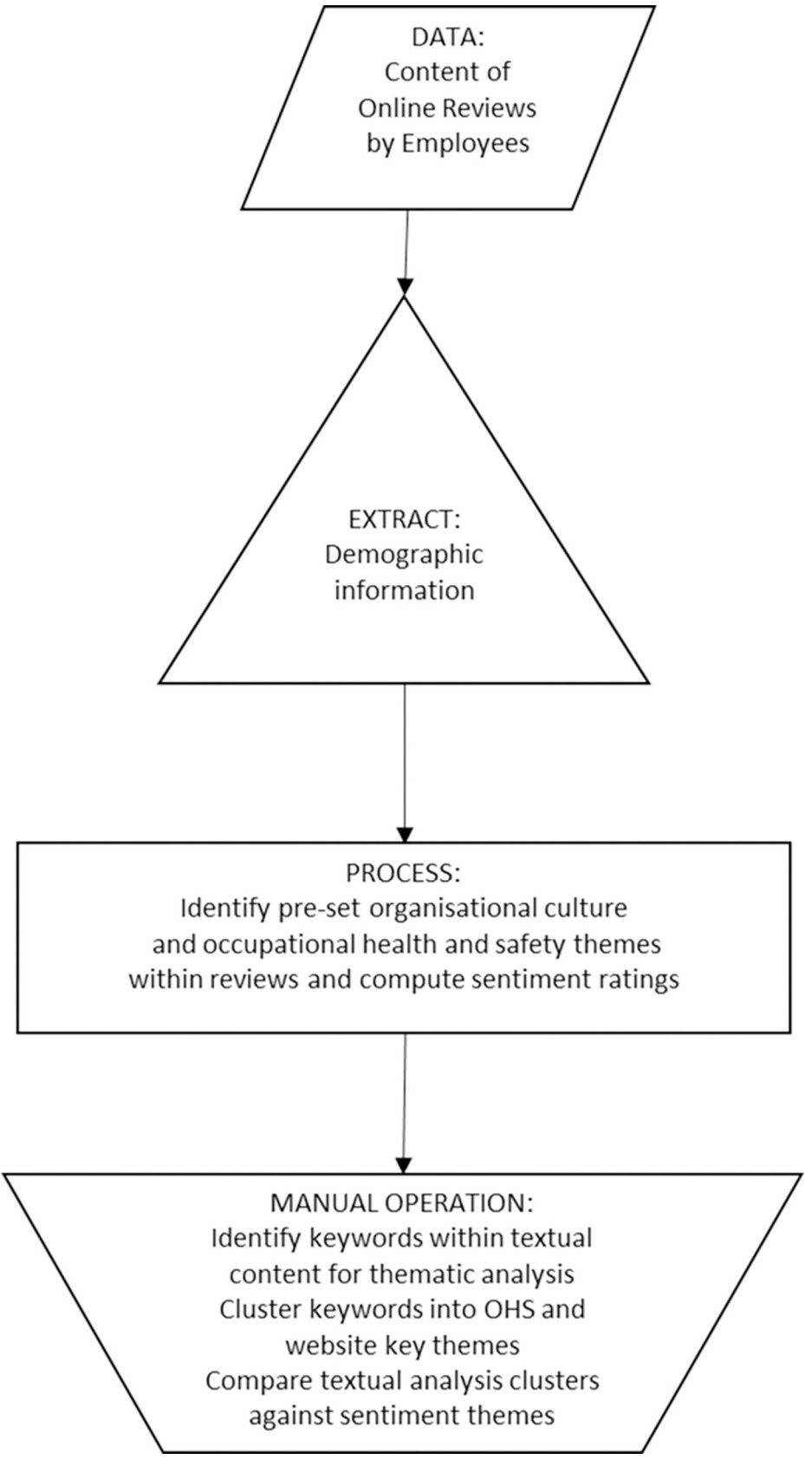

**Fig 1. Methodology for assessing and categorising content from hospital employee online reviews.**

**Table 3. Reviews across sites by number of current and former employees.**

| Source | Employment status | | | |
|---|---|---|---|---|
| | Unusable | Unclear / not indicated | Current | Former |
| Seek | 0 | 0 | 11 (39.3%) | 17 (60.7%) |
| Indeed | 0 | 1 (1.7%) | 21 (35.6%) | 37 (62.7%) |
| Glassdoor | 1 (6.3%) | 3 (18.7%) | 6 (37.5%) | 7 (43.8%) |
| | Usable Reviews | | | |
| Total | | 4 (3.9%) | 38 (37.3%) | 61 (59.8%) |

## Safety culture and occupational health and safety themes

Deductive thematic analysis of the textual content of the narrative section of the reviews was undertaken to categorise the reviews against the eight key Safety Attitudes dimensions (Fig 2). The definitions for these eight categories were derived from descriptions and instructions delineated within the Safety Attitudes Questionnaires employed by several hospitals to assess organisational climate and occupational health and safety amongst healthcare employees [6]. Stress recognition and risk management were mentioned only across seven and two reviews respectively, as these themes were not covered by the sites in their rating system. However, most online employee reviews addressed broad themes related to perceptions of management (n = 98), safety climate (n = 97), teamwork climate (n = 91) and working conditions (n = 98). A considerable number of reviews addressed themes related to job satisfaction (n = 49) and learning, training, and development (n = 41).

## Positive sentiment

The majority (72.8%; n = 75) of online employee reviews expressed positive sentiments. 33 positive reviews (32.3%) had been posted by current employees at the hospital network (Table 6). Several employees, both former and current, expressed organisational culture as being a key factor in contributing to their sense of satisfaction while working at the hospital. Some former employees even expressed regret at having had to quit their jobs at the hospital owing to external life circumstances. Positive statements about the organisation often included elements specific to workplace design, location, compensation, work-life balance, and a sense of community. Many reviews incorporated descriptions of the challenges of working in health-care, but these were not necessarily related to the particular hospital site or organisation. Online reviews acknowledged that some of the negative elements of working in the health sector that they had experienced were industry-specific rather than organisation-specific.

## Negative sentiment

Currency of employment with the hospital network was available for 73 of the employees who had posted online reviews (Table 7). Of these reviews, employee reviews expressing negative

**Table 4. Length of employment at hospital site indicated by reviewers across sites.**

| Source | Length of employment at site | | | | |
|---|---|---|---|---|---|
| | ≤1 year | 1–3 years | 3–5 years | 5–8 years | Unclear / not indicated |
| Seek | 6 (21.4%) | 8 (28.6%) | 8 (28.6%) | 2 (7.1%) | 4 (14.3%) |
| Indeed | | 1 (1.7%) | | 1 (1.7%) | 57 (96.6%) |
| Glassdoor | 1 (6.2%) | 3 (18.7%) | 5 (31.2%) | 2 (12.5%) | 5 (31.2%) |
| | Usable reviews | | | | |
| Total | 7 (6.8%) | 12 (11.7%) | 13 (12.7%) | 5 (4.9%) | 66 (64.7%) |

**Table 5. Role type indicated by reviewers across sites.**

| Source | Role | | |
|---|---|---|---|
| | Clinical* | Non-clinical+ | Unclear / not indicated |
| Seek | 13 (46.4%) | 14 (50.0%) | 1 (3.6%) |
| Indeed | 28 (47.4%) | 31 (52.5%) | 0 |
| Glassdoor | 8 (50.0%) | 5 (31.2%) | 3 (18.7%) |
| Usable reviews | | | |
| Total | 49 (48.0%) | 50 (49.0%) | 4 (3.9%) |

*Clinical roles–medical, nursing, and allied health

+Non-clinical roles–management, administrative and support services

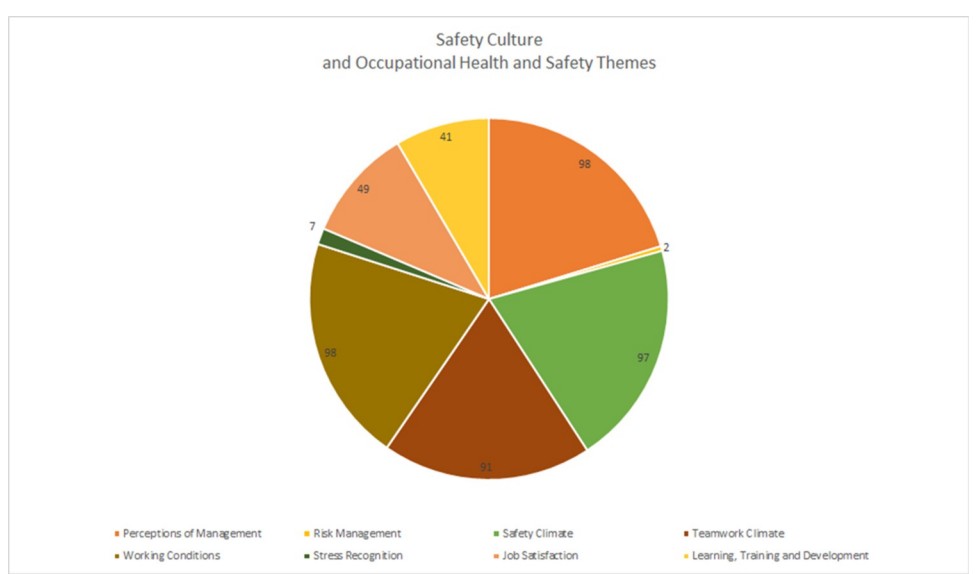

**Fig 2. Distribution of key themes from employee online reviews (n = 103).**

**Table 6. Sentiment distributions of reviews across sites.**

| Source | Number of reviews | Sentiment | | |
|---|---|---|---|---|
| | | Positive | Neutral | Negative |
| Seek | 28 (27.5%) | 19 (67.9%) | 3 (10.7%) | 6 (21.4%) |
| Indeed | 59 (57.8%) | 47 (79.6%) | 3 (5.1%) | 9 (15.3%) |
| Glassdoor | 16 (15.7%) | 9 (56.3%) | 3 (18.7%) | 3 (18.7%) |
| Usable Reviews | | | | |
| Total | 102 | 75 (73.5%) | 9 (8.8%) | 18 (17.7%) |

**Table 7. Online staff review sentiment by former employees and current employees.**

| Employment currency | Current employee | Former employee | Total |
|---|---|---|---|
| Review sentiment | | | |
| Negative | 2 (5.26%) | 16 (26.67%) | 18 (24.66%) |
| Neutral | 3 (7.89%) | 4 (6.67%) | 7 (9.59%) |
| Positive | 33 (86.84%) | 40 (66.67%) | 73 (100%) |

sentiments (n = 18) were largely posted by former employees (n = 16) and indicated areas of discontent that were related to organisational and systemic factors.

Reviews by former employees appeared to be used as a channel to express discontent once employees had exited the organization. Negative reviews overwhelmingly invoked themes of organisational values, morals, and justice. There was insufficient data to determine whether there were relationships between length of employment and type of employment (contractual, permanent, etc.) with the employee's reported experience. While online reviewers who expressed discontent mentioned organisational factors comprising culture, values, and management, dissatisfaction related to career opportunities, material compensation concerns, work-life balance and insufficient learning opportunities were mentioned as factors that negatively influence their views.

## Discussion

Safety culture within hospitals as demonstrated by survey assessment categories appears to be framed as distinct from employee satisfaction themes within operational measures of performance and organisational culture. Occupational health and safety frameworks within hospitals address themes ranging from top management support, safety systems, safety attitudes of staff, reporting incidents, communication openness, organizational learning, and teamwork to risk management and stress recognition [6, 13, 14]. While these themes measure the operational success of occupational health and safety, they may not measure the factors that hospital employees consider just as integral to their health, safety, and performance. Further, themes that are considered integral to organisational safety climate such as risk management and stress recognition were rarely mentioned by employees within online reviews. Themes such as work- life balance, organisational culture and values, career and advancement opportunities, and compensation and benefits appear to have a significant impact on hospital employees' sense of well-being, engagement, satisfaction, performance, and positive intention to continue working with the organisation. These factors are also likely to influence how employees speak of the organisation in public forums thereby potentially impacting the organisation's reputation as an employer. One online reviewer who had provided identifiable details about a current employee at the organisation appeared to have been contacted by the website to moderate their review. The details provided spanned another employee's identity, role, and professional associations. The risk of professional liability for both the website as well as the employer because of reviews such as these may be a deterrent to organisational support for engaging with the content of job websites. However, this instance appeared to have been an exception, and had been actively rectified by the website. Further, the anonymity of individuals on job websites did not appear to encourage malicious or overt negative sentiment. In fact, multiple reviewers had reflected on the inherent nature of their professions and the wider pressures facing hospitals while reviews with neutral or positive ratings.

## Conclusion

Online reviews are not designed to explicitly provide an assessment of organisational culture. Nevertheless, while this source of information may not provide the types of information sought from employees via safety assessment surveys within hospitals, they may offer a valuable source of information related to organisational culture and employee satisfaction. For instance, online employee reviews indicate that perceptions of positive employee satisfaction, engagement, and performance could be improved by enhancing opportunities for learning and development and achieving a work-life balance. While these findings have been reported through the results of traditional employee surveys [15], online reviews may serve as a

comparable source to gather this type of information from employees. Thus, this novel source of information could also be used to improve reputation management efforts and integrate employee well-being factors into human resource management efforts at hospitals. Factors that are considered significant by employees and that impact their well-being and performance may need to be integrated into organisational safety culture implementation and operational measures.

## Limitations

Online reviews carry an inherent risk of adulteration or mis-posting of reviews intended for another organisation with an identical name. Within our dataset, we encountered one such review referring to an American hospital. Unless there is a larger uptake of online reviews, the usage and therefore corpus of data may remain small for specific organisations and regions. This may limit the potential to derive insights for specific organisation or health networks. The sample and findings reported within this study are small and therefore, not representative of all hospitals, and cannot be extrapolated to apply beyond the network of hospitals considered for this study. Demographic details gleaned from online reviews cannot be elaborated in the same manner that traditional employee feedback instruments afford. Patient perceptions of physician quality have been found to reflect the opinions of a "vocal majority" over a silent majority [16]. There is a possibility that this risk might also apply to online reviews posted by hospital staff. In addition, the risk of self-selection that exists among survey respondents [17] could also be extended to online reviews. Nonetheless, the effectiveness of mean ratings to measure the relative quality of products or services being rated within online reviews has been found to be reliable for other industries and businesses [18]. Retrospective analysis undertaken in this study carries inherent limitations as employment status at the time of an individual posting a review may limit the usability of information provided through that review. Therefore, this methodology requires further development and rigorous validation against larger datasets. It is not yet suitable to replace traditional assessment models for organisational culture measurement.

## Implications

Healthcare organisational policy and management practitioners could use freely available content from online job websites to understand factors that may impact employee satisfaction, engagement, and performance within their hospitals. This source of information could also be used to improve reputation management efforts, and to provide insights to improve employee well-being, hiring and human resource management practices within healthcare organisations. Subject to further development and validation, this study contributes towards the foundational efforts in building context-aware machine learning algorithms and healthcare-sector-specific ontologies that elaborate hospital employee experience [19, 20]. These efforts could help standardise approaches to assessing, implementing, and evaluating organisational culture improvement initiatives across hospitals in the future.

## Acknowledgments

We acknowledge and thank Dr Rachel Urwin at the Centre for Health Systems and Safety Research (CHSSR) for her analysis of organisational survey instruments being used at hospital networks in Sydney.

## Author Contributions

**Conceptualization:** Antoinette Pavithra.

**Data curation:** Antoinette Pavithra.

**Formal analysis:** Antoinette Pavithra, Johanna Westbrook.

**Funding acquisition:** Antoinette Pavithra.

**Investigation:** Antoinette Pavithra, Johanna Westbrook.

**Methodology:** Antoinette Pavithra.

**Project administration:** Antoinette Pavithra, Johanna Westbrook.

**Resources:** Antoinette Pavithra.

**Supervision:** Johanna Westbrook.

**Visualization:** Antoinette Pavithra.

**Writing – original draft:** Antoinette Pavithra.

**Writing – review & editing:** Johanna Westbrook.

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
