## [Decision Letter · Decision Letter 0]

21 Feb 2022

PONE-D-21-07002An Assessment of Organisational Culture in Australian Hospitals Using Employee Online ReviewsPLOS ONE

Dear Dr. Pavithra,

Thank you for submitting your manuscript to PLOS ONE. After careful consideration, we feel that it has merit but does not fully meet PLOS ONE’s publication criteria as it currently stands. Therefore, we invite you to submit a revised version of the manuscript that addresses the points raised during the review process.

ACADEMIC EDITOR: Please insert comments here and delete this placeholder text when finished. Be sure to:Indicate which changes you require for acceptance versus which changes you recommendAddress any conflicts between the reviews so that it's clear which advice the authors should followProvide specific feedback from your evaluation of the manuscriptPlease ensure that your decision is justified on PLOS ONE’s publication criteria and not, for example, on novelty or perceived impact.

We look forward to receiving your revised manuscript.

Kind regards,

Oathokwa Nkomazana, MD MSC PhD

Academic Editor

PLOS ONE

Journal Requirements:

Additional Editor Comments (if provided):

Thank you for an interesting study. Please note comment from Reviewer 3; especially about the vocal minority eclipsing the silent majority. Also note comments by Reviewer 2 about the methods.

Reviewers' comments:

Reviewer's Responses to Questions

**Comments to the Author**

1. Is the manuscript technically sound, and do the data support the conclusions?

Reviewer #1: Yes

Reviewer #2: Partly

Reviewer #3: Yes

2. Has the statistical analysis been performed appropriately and rigorously? 

Reviewer #1: N/A

Reviewer #2: Yes

Reviewer #3: Yes

3. Have the authors made all data underlying the findings in their manuscript fully available?

Reviewer #1: No

Reviewer #2: No

Reviewer #3: Yes

4. Is the manuscript presented in an intelligible fashion and written in standard English?

Reviewer #1: Yes

Reviewer #2: Yes

Reviewer #3: Yes

5. Review Comments to the Author

Reviewer #1: The aim of this study was to analyse content of reviews posted by hospital employees on three job websites. It is understandable that performing rigorous statistics might have been a challenge given the small samples. Organizational culture is a complex concept, this study provides a baseline on which future studies could build on. Did the authors considered qualitative content analysis regarding trustworthiness?

Reviewer #2: There is a lot of details that needed to be provided to strengthen this study. A lot of statements across the study sections introduce ideas that are not well expanded or tied together. The methods section is confusing and has gaps that if revised could help improve the study, including clarity on how and why 103 reviews were extracted for the study period – sample size determination? how was data abstracted from the reviews? Providing the online review (i.e. details of the elements captured by these surveys) is also a critical addition that could be made to help provide clarity on study results and discussions. I find this study to be technically lacking with insufficient depth in the analysis and presentation of the results – the insufficient introduction also adds to this limitation.

Reviewer #3: The article is well written, methodologically sound, and relevant as it seeks to answer an important question: with the proliferation of online reviews of health care providers and health facilities, how good are these reviews at providing information related to organizational culture? The answer is that they can be quite good, with some caveats. My comments are pretty minor, and the authors should feel free to accept them or not. I thought it would be interesting to have a table that compares data/findings between former employees and current employees. I also through that in the discussion it might be worth expanding upon the risk that, with online reviews, the opinions of a "vocal" minority might end up overwhelming those of a quite majority

6. PLOS authors have the option to publish the peer review history of their article (what does this mean?). If published, this will include your full peer review and any attached files.

Reviewer #1: No

Reviewer #2: No

Reviewer #3: No

---

## [Author Response · Author response to Decision Letter 0]

1 Aug 2022

Response to reviewer #1:

Thank you for your comments.

Yes, we did consider qualitative content analysis. However, we have limited the qualitative thematic content analysis to deductive coding to assess whether the content of the reviews met the criteria across traditional assessment instruments and tools. So as to not infringe on copyrighted content by reproducing snippets of content from the online reviews, we did not perform inductive analysis, as we are not allowed to reproduce user reviews or extracts from these reviews within the article. 

Due to the general nature of the comments, and restrictions resulting from the global pandemic, follow-up interviews with hospital staff to verify the trustworthiness of these reviews was not possible at this time. We have reflected on the limitations arising from using online reviews including the risk of bias in these reviews in lines 326-332.

Response to reviewer #2: We have updated the methods section to specify that this set of 103 reviews comprised all reviews posted by users that were intelligible, referred to the right organisation and had been moderated and posted by the specified websites. (Additions made: lines 133-135)

Data were abstracted by reading the reviews online and abstracting information as described in the methods section (p 6-7)

We cannot reproduce the surveys used on each website due to copyright restrictions. However, we have described the rating systems offered by each site and how the relevant categories were used for thematic analysis between lines 139-190.

Thank you for your inputs in helping us revise our article for improved clarity. We have added additional references and information in the introduction to elaborate on key ideas addressed in the article, between lines 81-92, 95-99 and 103-109. 

Response to reviewer #3: Thank you for your comments. We have revised our manuscript to include your suggestions.

We have included a table presenting information around the currency of employment for employees who had posted online reviews where this information was available. This has been presented in lines 259-260 and 264-266.

We have also expanded our reflection about the self-selection bias and how this could arise from a vocal minority. We have referenced relevant literature to support the additions to our manuscript (lines 334-341).

---

## [Editor Report · Decision Letter 1]

23 Aug 2022

An Assessment of Organisational Culture in Australian Hospitals Using Employee Online Reviews

PONE-D-21-07002R1

Dear Dr. Pavithra,

We’re pleased to inform you that your manuscript has been judged scientifically suitable for publication and will be formally accepted for publication once it meets all outstanding technical requirements.

Kind regards,

Oathokwa Nkomazana, MD MSC PhD

Academic Editor

PLOS ONE
---

## [Editor Report · Acceptance letter]

5 Sep 2022

PONE-D-21-07002R1 

An Assessment of Organisational Culture in Australian Hospitals Using Employee Online Reviews 

Dear Dr. Pavithra:

I'm pleased to inform you that your manuscript has been deemed suitable for publication in PLOS ONE. Congratulations! Your manuscript is now with our production department. 

Kind regards, 

on behalf of

Dr. Oathokwa Nkomazana 

Academic Editor

PLOS ONE